# PARETO RANK-PRESERVING SUPERNETWORK FOR HW-NAS

## ABSTRACT

In neural architecture search (NAS), training every sampled architecture is very time-consuming and should be avoided. Weight-sharing is a promising solution to speed up the evaluation process. However, a sampled subnetwork is not guaranteed to be estimated precisely unless a complete individual training process is done. Additionally, practical deep learning engineering processes require incorporating realistic hardware-performance metrics into the NAS evaluation process, also known as hardware-aware NAS (HW-NAS). HW-NAS results in a Pareto front, a set of all architectures that optimize conflicting objectives, i.e. task-specific performance and hardware efficiency. This paper proposes a supernetwork training methodology that preserves the Pareto ranking between its different subnetworks resulting in more efficient and accurate neural networks for a variety of hardware platforms. The results show a 97% near Pareto front approximation in less than 2 GPU days of search, which provides x2 speed up compared to state-of-the-art methods. We validate our methodology on NAS-Bench-201, DARTS and ImageNet. Our optimal model achieves 77.2% accuracy (+1.7% compared to baseline) with an inference time of 3.68ms on Edge GPU for ImageNet.

## 1 INTRODUCTION

A key element in solving real-world deep learning (DL) problems is the optimal selection of the sequence of operations and their hyperparameters, called *DL architecture*. Neural architecture search (NAS) (Santra et al. (2021)) automates the design of DL architectures by searching for the best architecture within a set of possible architectures, called *search space*. When considering hardware constraints, hardware-aware neural architecture search (Benmeziane et al. (2021); Sekanina (2021)) (HW-NAS) simultaneously optimizes the task-specific performance, such as accuracy, and the hardware efficiency computed by the latency, energy consumption, memory occupancy, and chip area. HW-NAS works (Cai et al. (2019); Lin et al. (2021); Wang et al. (2022)) showed the usefulness and discovered state-of-the-art architectures for Image Classification (Lin et al. (2021)), Object detection (Chen et al. (2019)), and Keyword spotting (Busia et al. (2022)).

HW-NAS is cast as a multi-objective optimization problem. Techniques for HW-NAS span evolutionary search, Bayesian optimization, reinforcement learning and gradient-based methods. These require evaluating each sampled architecture on the targeted task and hardware platform. However, the evaluation is extremely time-consuming, especially for task-specific performance, which requires training in the architecture. Many estimation strategies (White et al. (2021)) are used to alleviate this problem, such as neural predictor methods (Benmeziane et al. (2022a); Ning et al. (2020)), zero-cost learning (Lopes et al. (2021); Abdelfattah et al. (2021)), and weight sharing (Chu et al. (2021); Chen et al. (2021)). These strategies are evaluated on how well they respect the ground truth ranking between the architectures in the search space.

Weight sharing is an estimation strategy that formulates the search space into a supernetwork. A supernetwork is an over-parameterized architecture where each path can be sampled. At the end of this sampling, a sub-network of the supernetwork is obtained. In each layer, all possible operations are trained. With this definition, we can classify weight-sharing NAS in two categories: (1)**a two-stage NAS** in which we first train the supernetwork on the targeted task. Then, using the pre-trained supernetwork, each sampled sub-network's performance can be estimated using a search strategy, such as an evolutionary algorithm. (2) **a one-stage NAS** in which we simultaneously search and

train the supernetwork. Additional parameters are assigned to each possible operation per layer. These parameters are trained to select which operation is appropriate for each layer.

Both Weight-sharing categories assume that the rank between different sub-networks is preserved. Two architectures with the same rank imply that they have the same accuracy. State-of-the-art works (Zhang et al. (2020); Peng et al. (2021); Zhao et al. (2021)) have highlighted the training inefficiency in this approach by computing the ranking correlation between the architectures' actual rankings and the estimated rankings. Some solutions have been proposed to train the supernetwork with strict constraints on fairness to preserve the ranking for accuracy, such as FairNAS (Chu et al. (2021)). Others train a graph convolutional network in parallel to fit the performance of sampled sub-networks Chen et al. (2021). However, current solutions have two main drawbacks:

1. In the multi-objective context of HW-NAS, different objectives such as accuracy and latency have to be estimated. The result is a Pareto front, a set of architectures that better respects the trade-off between the conflicting objectives. The ranking following one objective is no longer a good metric for the estimator. In this setting, we need to take into account the dominance concept in the ranking. Both estimations hinder the final Pareto front approximation and affect the search exploration when considering the accuracy and latency as objectives.

2. Many works (Chen et al. (2021); Zhao et al. (2021); Guo et al. (2020)) attempt to fix the supernetwork sampling after its training. We believe that this strategy is inefficient due to the pre-training of supernetwork. Its accuracy-based ranking correlation is bad. In Dong & Yang (2020), a reduced Kendall's tau-b rank correlation coefficient of 0.47 has been obtained on NAS-Bench-201 when using this approach. The accuracy estimation is thus non-conclusive and will mislead any NAS search strategy.

To overcome the aforementioned issues, we propose a new training methodology for supernetworks to preserve the Pareto ranking of sub-networks in HW-NAS and avoid additional ranking correction steps. The contributions of this paper are summarized as follows:

- We define the **Pareto ranking as a novel metric** to compare HW-NAS evaluator in the multi-objective context. Our study shows that optimizing this metric while training the supernetwork increases the Kendall rank correlation coefficient from 0.47 to 0.97 for a Vanilla Weight-sharing NAS.

- We introduce a novel **one-stage weight-sharing supernetwork training methodology**. The training optimizes the task-specific loss function (e.g. cross-entropy loss) and a Pareto ranking listwise loss function to select the adequate operation per layer accurately.

- During training, **we prune the operations that are the least likely to be in the architecture of the optimal Pareto front**. The pruning is done by overlapping the worst Pareto-ranked sub-networks and removing the operations that are only used in these sub-networks.

We demonstrate that using our methodology on three different search spaces, namely NAS-Bench-201 (Dong & Yang (2020)), DARTS (Liu et al. (2019)) and ProxylessNAS search space (Cai et al. (2019)), we achieve a higher Pareto front approximation compared to current state-of-the-art methods. For example, we obtained 97% Pareto front approximation when One-Shot- NAS-GCN (Chen et al. (2021)) depicts only 87% on NAS-Bench-201.

## 2    BACKGROUND & RELATED WORK

This section summarizes the state-of-the-art in accelerating multi-objective optimization HW-NAS.

### 2.1    ACCELERATING HARDWARE-AWARE NAS

Given a target hardware platform and a DL task, **Hardware-aware Neural Architecture Search (HW-NAS)** (Benmeziane et al. (2021)) automates the design of efficient DL architectures. HW-NAS is a multi-objective optimization problem where different and contradictory objectives, such as accuracy, latency, energy consumption, memory occupancy, and chip area, have to be optimized. HW-NAS has three main components: (1) *the search space* ,(2) *the evaluation method* and (3) *the*

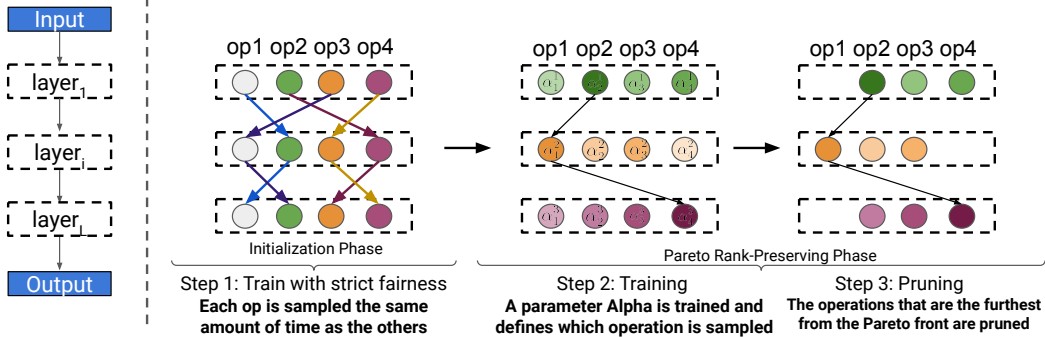

Figure 1: Our Pareto Rank-Preserving Training methodology for Supernetwork. The strongest shades illustrate the most important operations for each layer at each iteration. $\alpha_o^l$ corresponds to the parameter alpha associated with layer $l$, operation $o$.

*search strategy* The main time-consuming component in HW-NAS is the evaluation method. Several state-of-the-art works (White et al. (2021)) have been proposed to alleviate this problem.

**Predictor-based methods** (Ning et al. (2020); Lomurno et al. (2021)) are the most popular strategies where machine learning models are used to predict the accuracy or latency from the architecture features (e.g. number of convolutions, widening factor, etc.) or its representation using Graph Neural Networks (GNN) (Ning et al. (2020)) and Recurrent Neural Networks (RNN) (Lomurno et al. (2021)). However, these methods are not flexible to different search spaces as they require training a sampled dataset and then training the predictor.

**Weight-sharing approaches** (Chu et al. (2021); Chen et al. (2021); Zhao et al. (2021); Guo et al. (2020)), on the other hand, define the search space as a supernetwork. In each layer, the supernetwork combines the results of possible operations. A sequence of operations from the input to the output is called a sub-network and constitutes a possible architecture. Training the supernetwork consists of training several paths at once. The input is forwarded through a series of parallel operations whose outputs are summed after each layer. There are two main issues when training a supernetwork:

1. **The order of the sampled sub-networks matters:** Assume we have two sub-networks $A$ and $B$. Both $A$ and $B$ start with the same operation $op_1$ in layer 1. During the first training iteration, $A$ is sampled and $op_1$ weights are adjusted. The second iteration samples $B$ and adjusts $op_1$ weights again. If we want to evaluate $A$, we would use the new adjusted weights of $op_1$ which degrades the estimation.

2. **Unfair Bias:** Sub-networks with an initial better task-specific performance are more likely to be sampled next and maintain a higher coefficient in one-stage supernetwork. Fairnas (Chu et al. (2021)) defines strict fairness constraints that ensure that each operation's weights are updated the same amount of times at each stage.

## 2.2 MULTI-OBJECTIVE OPTIMIZATION IN HW-NAS

Optimizing conflicting objectives simultaneously requires the definition of a decision metric. In multi-objective optimization Batista et al. (2011), this metric is the dominance criteria. In a two-objectives minimization problem, dominance is defined as: *If $s_1$ and $s_2$ denote two solutions, $s_1$ **dominates** $s_2$ ($s_1 \succ s_2$) if and only if $\forall i \; f_i(s_1) \leq f_i(s_2)$ AND $\exists j \; f_j(s_1) < f_j(s_2)$. $f_i$ and $f_j$ are* conflicting objective functions such as latency and accuracy.

Using the dominance, there is no single solution that dominates all the others. We instead build the *Pareto front*; the set of all dominant solutions. The Pareto front approximation is evaluated using the hypervolume metric. The hypervolume measures the area dominated by a Pareto front approximation $P$ and a reference point. The reference point is defined as an architecture with a high

latency and low accuracy (furthest from the optimal points). The maximization of hypervolume leads to a high-qualified and diverse Pareto front approximation set.

In HW-NAS, computing the hardware efficiency is expensive due to the time-consuming deployment and measurements on the hardware. Using multiple performance estimators is thus popular Hu et al. (2019); Elsken et al. (2019); Lu et al. (2020); Huang & Chu (2021). Current multi-objective HW-NAS approaches focus on optimizing the search algorithm at the expense of poor performance estimators. However, using a performance estimator per objective is not optimal Benmeziane et al. (2022b). In this paper, we present an original weight-sharing technique that directly predicts a multi-objective metric, called *Pareto ranking*.

## 3 METHODS

The core motivation for a novel training methodology is to achieve an efficient sub-networks evaluation for HW-NAS. The proposed training methodology must preserve the Pareto ranking between different sub-networks while reducing the overall search time.

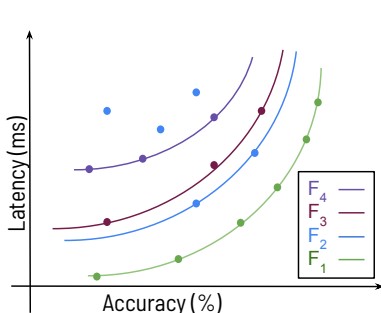

Figure 2: Illustration of the Pareto rank and $F_i$

Figure 3: Supernetwork definition when coupling task-specific weights $W$ and operation's score parameters $\alpha$. *Conv 3x3* is the operation with the highest selection score.

### 3.1 PARETO RANKING

In this section, we define the Pareto ranking metric used to train and evaluate the supernetwork.

*Pareto Ranking* Solving the multi-objective optimization problem on a set of sub-networks results in a Pareto front. This set of architectures in this front is denoted as $F_1$, i.e., all the architectures have a rank of 1. We achieve the lower ranks by successfully solving the problem on the set of sub-networks pruned from the previous solutions. The lowest rank is assigned to the sub-networks that do not dominate any sub-network. We formally define the Pareto ranking in equation 1, where $S$ is the entire supernetwork, $F_{k'}$ is a set of sub-networks ranked $k'$, and $\succ$ is the dominance operation.

Using this ranking, multiple architectures may have the same rank. This happens when none of them can dominate the others.

$$\text{a is ranked k} \iff \forall \hat{a} \in S - \bigcup_{s_i \in F_{k'} \wedge k' < k} , \hat{a} \succ a \tag{1}$$

***Pareto Ranking Correlation.*** We evaluate the quality of an estimator using ranking correlations such as Kendall's tau-b Correlation or Spearman Correlation. Kendall's tau-b determines whether there is a monotonic relationship between two variables and is suitable when variables contain many

tied ranks Benmeziane et al. (2021), which is our case. In the rest of the paper, we compute Kendall's Tau-b correlation between the ground truth ranks (i.e. the Pareto ranks obtained from independently training the sub-networks), and the Pareto ranks obtained by evaluating each architecture with the supernetwork shared weights.

## 3.2 PARETO RANK-PRESERVING TRAINING

Our training methodology aims at preserving the Pareto ranking obtained by the weight-sharing evaluation.

Figure 3 shows a representation of the supernetwork definition and the different parameters we aim to learn. A sub-network is a path from the input to the output. All extracted sub-networks are of the same depth. We train the supernetwork with two goals: 1) enhance the task-specific loss function by adjusting $W$, the task-specific weights of the original model associated with the neural network operations such as the kernels in convolution, and 2) improve the Pareto ranking loss between its different paths by adjusting $\alpha$, the weights associated with the operation selection. $\alpha$ measures which operation is critical and which one is selected.

Algorithm 1 and figure 1 summarize the training procedure.

- **Step 1: Train with Strict Fairness** We train our supernetwork using FairNAS (Chu et al. (2021)) strict fairness constraint. This step adjusts the weights of all the sub-networks $W$ and gives a good starting point for the Pareto ranking training. Additionally, the accuracy estimation on the task-specific loss at this point is well estimated. We use these estimations to compute the true Pareto ranks in case no accuracy was provided by the benchmark.

- **Step 2: Pareto ranking training** For each iteration, we apply:

  **- Training to solve the task:** A mini-batch is sampled from the training set, and a sub-network is chosen according to each operation's highest $\alpha$. The operation's weights are updated using the task-specific loss, e.g., cross-entropy loss for image classification.

  **- Pareto rank training:** In this phase, we purposefully bias the training towards better Pareto-ranked architectures using the $\alpha$ parameters. $\alpha$ parameters are trained using the loss function provided in equation 2. During the forward pass, we Pareto rank the sampled sub-networks. We compute the number of times an operation $op_i$ appears in layer $l_j$ on $N$ top-ranked sub-networks, denoted as $g(op_i, l_j)$. $N$ is a hyperparameter defined before training. We denote by $\hat{g}(op_i, l_j)$, the ground truth. Equation 2 computes the hinge loss over all layers in the sampled sub-networks and compares the number of times the operation with the highest $\alpha$ appears in the predicted Pareto front and the ground truth one.

$$L = \sum_{j=1}^{L} \sum_{i, g(op_i, l_j) > \hat{g}(op_i, l_j), i \neq \arg\max(\alpha)} max[0, m - g(\arg\max(\alpha), l_j) - \hat{g}(op_i, l_j)] \quad (2)$$

  We adjust each operation's $\alpha$ parameters and compute each sampled sub-network's latency using a lookup table. We define the predicted Pareto score according to $P_s = \sum_{op \in a} \alpha_{op}$, i.e., the sum of selected operations' alpha values. Next, we compute the listwise ranking loss defined by the cross entropy between the ranking scores and the Pareto ranks (ground truth).

- **Step 3: Pruning by Pareto Ranking Sub-networks** We drop sub-networks furthest from the optimal Pareto front to accelerate the training. First, we select the sub-networks belonging to the two first Pareto ranks. Then, based on the hypervolume improvement (HVI) (Emmerich et al. (2011)), we select $n$ sub-networks. The operations never used by any sub-network in this selection are removed for each layer. Equation 3 illustrates how the hypervolume improvement is computed in this context. $o_{ij}$ denotes operation $i$ in layer $j$. $HV$ denotes the hypervolume function and $\{S_{o_{ij}}\}$ denotes the set of sampled sub-networks using operation $i$ in layer $j$.

$$HVI(o_{ij}, P) = HV(P \bigcup \{S_{o_{ij}}\}) - HV(P - \{S_{o_{ij}}\}) \quad (3)$$

Finally, going over all the layers to select the operations with the highest $\alpha$ would suffice to find the most efficient DNN within the search space.

Figure 4 shows the training results. We compare our methodology to FairNAS (Chu et al. (2021)) strict fairness training. During training, the Pareto ranking correlation increases with the quality of the estimations. When using our training methodology without considering the alpha parameters, the ranking correlation saturates at 0.7. FairNAS achieves the same behaviour with reduced variance among the different training runs. However, if we consider the alpha parameters, the selection is more efficient and the architectures' rankings are well represented with 0.94.

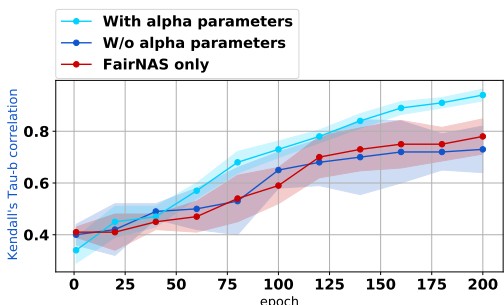

Figure 4: Training performance computed with the Kendall's Tau Correlation between the independently trained Pareto ranks and the estimated Pareto ranks obtained by training the supernetwork.

---

**Algorithm 1** Supernetwork Training Algorithm

**Input:** Search space $S$, number of epochs for fairness training $N_f$, number of epochs for Pareto training $N_p$, Supernetwork parameters $(W, \alpha)$, training dataloader $D$, task-specific loss $Loss$, Pareto raking loss $Loss_{PR}$, number of sampled sub-network $n$

**procedure** TRAIN
 Initialize $W$ and $\alpha$ for each operation in Supernetwork
 Strict fairness training for $N_f$ epochs
 **for** i=1 to $N_p$ **do**
  **for** data, labels in $D$ **do**
   Build $model$ with $argmax(\alpha)$ following step 2
   Reset gradients to zero for all $W$ parameters
   Calculate gradients based on $Loss$, data, labels and update $W$ by gradients
  **end for**
  Sample $n$ sub-networks, $models$
  Compute: Pareto rank of $models$, $Loss_{PR}$ between scores and Pareto rank.
  Update $\alpha$ by gradients
 **end for**
 **end procedure**

---

## 4 EXPERIMENTS

In this section, we evaluate our training methodology on three search spaces: NAS-Bench-201 (Dong & Yang (2020)), DARTS (Liu et al. (2019)) and ProxylessNAS Search space (Cai et al. (2019)).

### 4.1 SETUP

**Search Spaces:** Several search spaces have been used to evaluate our method's performance. NAS-Bench-201 (Dong & Yang (2020)) is a tabular benchmark that contains 15k convolutional neural networks. Each architecture is trained on CIFAR-10, CIFAR-100 and ImageNet-16-120 (Chrabaszcz et al. (2017)). We use the latency values obtained from HW-NAS-Bench (Li et al. (2021)). DARTS Liu et al. (2019) is a supernetwork benchmark that contains $10^{18}$ architectures. Each architecture is trained on CIFAR-10 and is transferable to ImageNet. We also validate our methodology on ImageNet using ProxylessNAS search space Cai et al. (2019) whose size goes to $6^{19}$. All training hyperparameters are listed in Table 5 in Appendix F.

| Architecture | Edge GPU | | | Mobile Phone : Pixel 3 | | | HW Aware | GPU Days |
| --- | --- | --- | --- | --- | --- | --- | --- | --- |
| | Top-1 Test Acc. | Params (M) | Latency (ms) | Top-1 Test Acc. | Params (M) | Latency (ms) | | |
| DARTS Liu et al. (2019) | 68.3 ± 0.08 | 3.4 | 5.36 | 68.3 ± 0.08 | 3.4 | 11.4 | No | 4 |
| ENAS Pham et al. (2018) | 53.89 ± 0.16 | 4.6 | 6.32 | 53.89 ± 0.16 | 4.6 | 19.8 | No | 0.16 |
| GDAS Dong & Yang (2019) | 90.89 ± 0.08 | 3.4 | 5.21 | 90.89 ± 0.08 | 3.4 | 10.36 | No | 0.21 |
| FairNAS Chu et al. (2021) | 93.23±0.18 | 3.2 | 4.68 | 92.4 ± 0.15 | 3.6 | 8.65 | Yes | 10 |
| PRP-NAS-BL (Ours) | 92.34 ± 0.05 | 3.0 | **2.3** | 89.54 ± 0.07 | 2.8 | **3.6** | Yes | 2 |
| PRP-NAS-BA (Ours) | **94.37 ± 0.02** | 4.5 | 4.35 | **94.2 ± 0.03** | 4.3 | 5.6 | Yes | 2 |
| PRP-NAS-O (Ours) | 93.65 ± 0.01 | 4.3 | 3.64 | 93.74 ± 0.00 | 3.4 | 4.61 | Yes | 2 |

Table 1: Comparison on NAS-Bench-201 CIFAR-10 on Edge GPU (Jetson Nano) and Mobile phone (Pixel 3).

## 4.2 SEARCH RESULTS

In these experiments, we consider two objectives: accuracy and latency (inference time). The latency is either given by HW-NAS-Bench (Li et al. (2021)), or computed using a lookup table as explained in section 3.

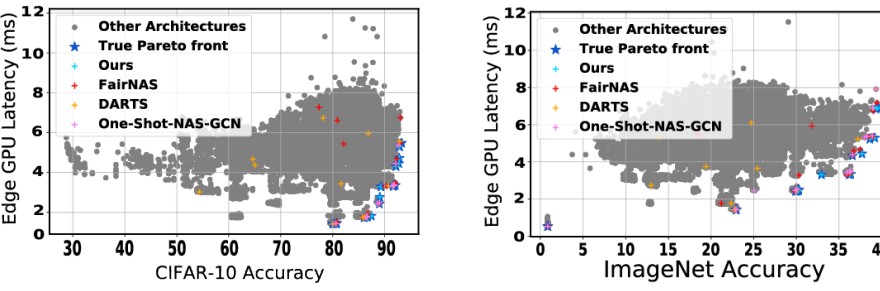

Figure 5: Pareto front approximation comparison on CIFAR-10 and ImageNet.

Figure 5 shows the Pareto front approximations obtained using different methods on NAS-Bench-201 for CIFAR-10 and ProxylessNAS Search space for ImageNet. We obtain a 10% hypervolume increase on NAS-Bench-201 and a 43% hypervolume increase on ImageNet compared to the best baselines: One-Shot-NAS-GCN and FairNAS, respectively.

### 4.2.1 SEARCH ON NAS-BENCH-201

Table 1 shows the results of our methodology on NAS-Bench-201 compared to state-of-the-art methods. PRP-NAS-BL, PRP-NAS-BA and PRP-NAS-O are three sampled architectures from our final Pareto front. BL stands for "Best Latency". BA stands for "Best Accuracy", and O stands for "Optimal". Notably, our architecture obtains highly competitive results. The optimal architecture, *PRP-NAS-O*, outperforms current state-of-the-art methods in accuracy and latency. Including hardware awareness during the search allows us to obtain flexible results according to the targeted hardware platform. Besides, multiple training runs show the stability of our method compared to other baselines. The acceleration in the search cost is mainly due to applying the pruning while training. This cost can vary according to the used GPU. We used GPU V100 to train the supernetwork. Results on other targeted platforms, can be found in Appendix B.

### 4.2.2 SEARCH ON IMAGENET

Similar conclusions can be extracted when searching on ImageNet. Table 2 summarizes the results. Our optimal model surpasses FairNAS-A (+1.9%) and One-Shot-NAS-GCN (+1.7%) while running faster. Training on Imagenet is time-consuming due to the difference in image resolution, which explains the increase in the search cost. We still surpass most of the methods in terms of search time. We compare two ProxylessNAS architectures; ProxylessNAS-R is specific to Mobile inference.

When using data augmentation and architecture tricks, namely Squeeze-and-excitation and AutoAugment, in the optimal architecture, we achieve 78.6% accuracy on Imagenet. However, this may affect the latency badly. On FPGA ZCU102, the latency increases from 4.63ms to 7.9ms.

### 4.3 RANKING QUALITY

The ranking preservation measures the quality of the evaluation component in NAS. In HW-NAS, we argue that this measure should consider the Pareto ranking instead of the independent ranks of each objective. We compare different estimators used in HW-NAS using Kendall's Tau Correlation between the predicted Pareto ranks and the Pareto ranks obtained from independently training the architectures. These latter are extracted from NAS-Bench-201. Figure 6 shows the correlation results. In general, it is more complex to train a supernetwork to respect the Pareto ranks because of the impact of the sub-networks on each other, i.e., the outputs of each layer are summed together. The increase in Kendall's tau correlation of previous weight-sharing methodology is due to the improvement in the accuracy estimation provided by the supernetwork.

Predictor-based evaluators use the learning-to-rank theory and train their predictors only to predict the ranking. Methods such as GATES (Ning et al. (2020)) or BRP-NAS (Dudziak et al. (2020)) train many independent predictors, one for each objective. HW-PR-NAS (Benmeziane et al. (2022a)) trains a single predictor to fit the Pareto ranks. However, their methodology is not flexible for supernetwork training.

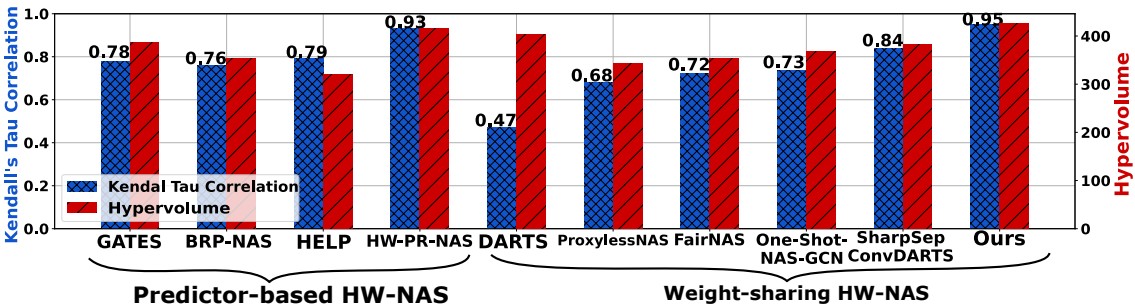

Figure 6: Kendall's Tau-b correlation and hypervolume comparison using different estimators on DARTS.

### 4.3.1 ANALYSIS OF $\alpha$ PARAMETER

Figure 7 illustrates the evolution of alpha parameters for each operation in layer 1 and 2 during the training. It clearly shows how alpha favors one operation over the others during training. At the end of the training, we take the operations with the highest alpha that represents the operations constructing architectures in the final Pareto front. If one layer has a clear candidate such as layer 1, with conv3x3 that exceeds 60%, this operation is then chosen. If a layer contains multiple operations with similar alpha values, we constructs all the path of that layer.

### 4.4 BATTERY USAGE PRESERVATION

The amount of energy consumed by each model can be different. It is mainly attributed to the number of multi-adds computed. We take supernetwork usage to another level by adequately scheduling the run of different sub-networks according to the system's battery life. In this experiment, the training is done with two objectives: accuracy and energy consumption. Once the training is done, only the

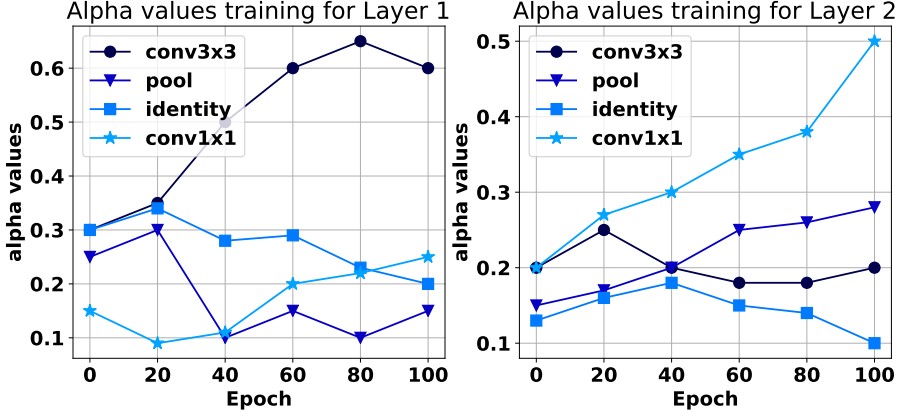

Figure 7: Analysis of trained alpha values for layer 1 and 2

Pareto front solutions are kept in the supernetwork, thanks to the pruning. We further select, from the final Pareto front, $s$ architectures. In this experiment $s = 5$. The total size of the supernetwork is then reduced to 20.5MB, comparable to MobileNet-V3 Large with 21.11MB. We deploy the model on a smartphone application that is always on. The application repeats the inference classification of one image. The application initially uses the sub-network with the highest accuracy. We switch to a lower accurate model every five hours for a better energy preserving. Figure 8 shows the results of the system's battery life while running the application for 24 hours. We use three scenarios:

1. **Worst Battery Usage:** From the Pareto front, we select the most accurate architecture. This is the only architecture the application runs and is the only one loaded in memory.

2. **Best Battery Usage:** Similar to the worst battery usage, we select the most energy-efficient.

3. **Adequate Battery Usage:** We load the complete supernetwork and switch the sub-network every 5 hours.

Using this strategy helps save up to 34% of the battery life while using highly accurate models most of the time. The average accuracy of the five selected sub-networks is 75.2%.

## 5    CONCLUSION

This work analyzes Hardware-aware weight-sharing NAS where the multi-objective context requires the estimator to preserve the Pareto rankings between sub-networks accurately. Contrary to standard baselines that independently estimate each objective, we propose a supernetwork training methodology able to preserve the Pareto rankings during the search. Using our methodology, we achieve  97% near Pareto front approximation on NAS-Bench-

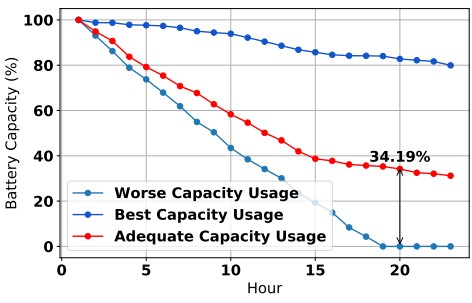

Figure 8: Battery life management.

201, DARTS, and ProxylessNAS Search Spaces. We find a 77.2% accuracy model on ImageNet while only training the supernetwork for 3.8 days. Using the supernetwork capabilities, we saved up to 34% of the battery capacity with an average accuracy of 75.2%.

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

## A    RESULTS ON IMAGENET

| Architecture | Edge GPU | | | Mobile Phone : Pixel 3 | | | HW Aware | GPU Days |
|---|---|---|---|---|---|---|---|---|
| | Top-1 Test Acc. | Params (M) | Latency (ms) | Top-1 Test Acc. | Params (M) | Latency (ms) | | |
| DARTS Liu et al. (2019) | 73.3 ± 0.03 | 4.7 | 5.36 | 73.3 ± 0.03 | 4.7 | 11.5 | No | 4 |
| ProxylessNAS Cai et al. (2019) | 75.1 ± 0.00 | 7.1 | 5.1 | - | - | - | Yes | 8.3 |
| ProxylessNAS-R Cai et al. (2019) | - | - | - | 74.6 | 4.0 | 6.8 | Yes | 8.3 |
| FairNAS-A Chu et al. (2021) | 75.3 ± 0.05 | 4.6 | 5.32 | 75.1 ± 0.06 | 4.3 | 6.84 | Yes | 12 |
| One-Shot-NAS-GCN Chen et al. (2021) | 75.5 ± 0.09 | 4.4 | 5.23 | 75.5 ± 0.09 | 4.4 | 9.4 | No | 4.7 |
| PRP-NAS-BL (Ours) | 68.95 ± 0.01 | 3.7 | 3.2 | 67.56 ± 0.05 | 3.1 | 4.8 | Yes | 3.8 |
| PRP-NAS-BA (Ours) | 77.5 ± 0.02 | 4.8 | 4.68 | 76.94 ± 0.02 | 4.4 | 6.13 | Yes | 3.8 |
| PRP-NAS-O (Ours) | 77.2 ± 0.03 | 4.6 | 3.68 | 75.6 ± 0.02 | 3.8 | 5.68 | Yes | 3.8 |

Table 2: Comparison on ImageNet on Edge GPU (Jetson Nano) and Mobile Phone (Pixel 3).

## B    ADDITIONAL RESULTS

Table 3 shows the results of our training methodology on FPGA ZCU 102 and Raspberry Pi3. Our methodology consistently outperforms state-of-the-art methods on different hardware platforms.

| Architecture | FPGA ZCU102 | | | Raspberry Pi 3 | | | HW Aware | GPU Days |
|---|---|---|---|---|---|---|---|---|
| | Top-1 Test Acc. | Params (M) | Latency (ms) | Top-1 Test Acc. | Params (M) | Latency (ms) | | |
| DARTS | 68.3 ± 0.08 | 3.4 | 7.32 | 68.3 ± 0.08 | 3.4 | 45.36 | No | 4 |
| ENAS | 53.89 ± 0.16 | 4.6 | 8.91 | 53.89 ± 0.16 | 4.6 | 35.8 | No | 0.16 |
| GDAS | 90.89 ± 0.08 | 3.4 | 4.98 | 90.89 ± 0.08 | 3.4 | 41.8 | No | 0.21 |
| FairNAS | 92.9±0.23 | 3.4 | 5.12 | 92.51 ± 0.9 | 3.3 | 34.15 | Yes | 10 |
| PRP-NAS-BL (Ours) | 91.35 ± 0.04 | 3.2 | 3.6 | 88.7 ± 0.03 | 2.4 | 7.6 | Yes | 2 |
| PRP-NAS-BA (Ours) | 94.37 ± 0.01 | 4.9 | 6.8 | 93.68 ± 0.05 | 4.68 | 40.7 | Yes | 2 |
| PRP-NAS-O (Ours) | 93.55 ± 0.04 | 4.2 | 4.23 | 92.54 ± 0.02 | 3.6 | 18.5 | Yes | 2 |

Table 3: Comparison to baselines on CIFAR-10 on FPGA ZCU-102 and Raspberry Pi3

## C    NUMBER OF SAMPLED SUB-NETWORKS

Figure 9 shows the effect of increasing the number of sampled sub-networks on the search results. Generally, increasing the number of samples, increases the hypervolume. The hypervolume is used to evaluate Pareto front approximations. It computes the area contained by the Pareto front points found by the search and a reference point. Our reference point is set as a pre-sampled architecture from the supernetwork, with a low accuracy and high latency. When the number of sampled sub-networks is too high, each layer's output is the sum of multiple operations that can or cannot be within the final Pareto front which induces a bias when adjusting the $alpha$ parameters.

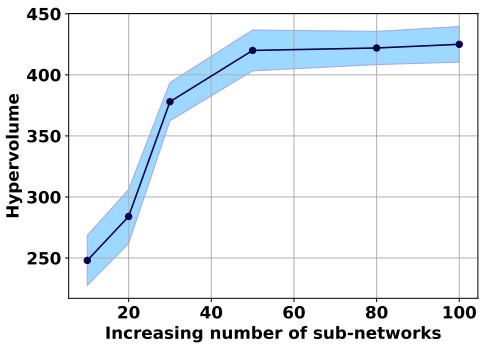

Figure 9: Hypervolume analysis with increasing number of sampled sub-networks for the final Pareto front throughout the search (higher is better) on NAS-Bench-201.

## D  PRUNING ALGORITHM

We validate the results of our pruning algorithm by comparing the results of our algorithm with and without it in table 4. Obviously without the pruning, the search time exponentially increases from 3.8 GPU days to 8.1. However, the hypervolume improves slightly. The final most accurate architecture is in both Pareto front obtained with and without pruning. The optimal architecture using pruning is better in terms of accuracy and latency. The latency is computed on Jetson Nano Edge GPU.

| Model | Test Acc (%) | Latency (ms) | Search Hypervolume | GPU days |
|---|---|---|---|---|
| PRP-NAS-O | 93.65 | 3.64 | 423.45 | 3.8 |
| PRP-NAS-O-no_pruning | 92.1 | 3.26 | 433.09 | 15.1 |

Table 4: Ablation results of Pruning of Pareto ranking.

## E  LATENCY ESTIMATION

In this section, we compare different latency estimators to validate the use of LUT during the search. We randomly extract 1000 architectures from NAS-Bench-201 and 1000 from DARTS. We measure the exact latency on Jetson Nano for each architecture. We train two predictor-based models, namely XGBoost and MLP with 3 layers. The training dataset contains 700 architectures and 300 were used for testing. On NAS-Bench-201, the architectures have a sequential execution which made LUT the most accurate in terms of latency ranking the architectures. On DARTS, XGBoost prediction was the most suitable methods. But, LUT was not far with 0.915 against 0.942. Computing the LUT in our algorithm is simple. Using a hook during the forward function on a PyTorch model is sufficient and much more direct than calling a surrogate model. We thus use this strategy to estimate the latency in our method.

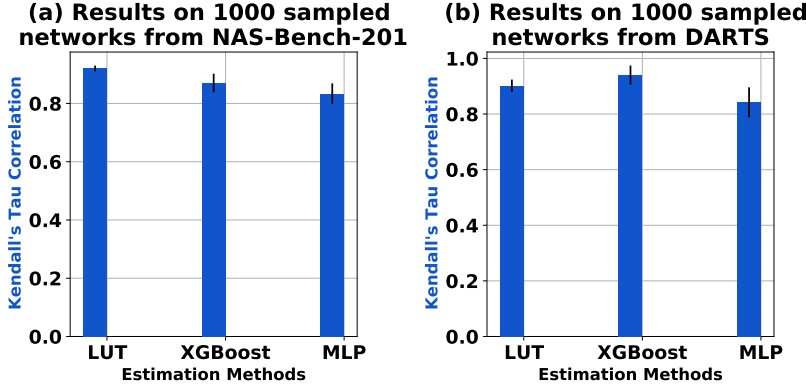

Figure 10: Comparison of latency estimators on Jetson Nano.

## F    TRAINING HYPERPARAMETERS

The training hyperparameters are listed in Table 5. It takes 2, 3.8, 3.8 GPU-days for NAS-Bench-201, DARTS and ProxylessNAS search space to train each supernetwork to fullness. Our training is 5x faster than previous works due to the pruning strategy. To be consistent with previous works, we do not employ data augmentation tricks such as cutout or mixup. We also do not employ any special operations such as squeeze-and-excitation. All these methods can further improve the scores on the test set.

| Benchmark | Hyperparameter | Value |
|---|---|---|
| NAS-Bench-201 | Nf | 20 |
| | Np | 50 |
| | n | 50 |
| | batch_size | 128 |
| | lr | 0.01 |
| | optim | SGD |
| | momentum | 0.9 |
| DARTS | Nf | 30 |
| | Np | 150 |
| | n | 100 |
| | batch_size | 256 |
| | lr | 0.025 |
| | optim | SGD |
| | momentum | 0.9 |
| ProxylessNAS Search Space | Nf | 30 |
| | Np | 150 |
| | n | 100 |
| | batch_size | 256 |
| | lr | 0.025 |
| | optim | SGD |
| | momentum | 0.9 |

Table 5: Training Hyperparameters

