# OpenReview forum: "Pareto Rank-Preserving Supernetwork for HW-NAS"
_ICLR.cc/2023/Conference — Submitted to ICLR 2023_

### Official Review · Reviewer_4mMw · 2022-10-21

**Confidence:** 5
**Correctness:** 3
**Technical Novelty And Significance:** 2
**Empirical Novelty And Significance:** 2
**Recommendation:** 3

**Clarity, Quality, Novelty And Reproducibility:**

Lots of important details of the proposed methods are missing, I can not reproduce the proposed method based on the given manuscript.

**Strength And Weaknesses:**

### Positive points:

- The authors introduce the Pareto ranking as a novel metric for multi-objective architecture search.

- The authors propose a supernetwork training strategy for preserving the Pareto ranking between different subnetworks.

 - The authors propose to prune the operations based on the optimal Pareto front while training the supernetwork.

### Negative points:

 - In the Pareto rank-preserving training section, the details of the proposed method are missing. In other word, after reading this paper, I can not re-implement the proposed method. It would better for the authors to make it more clear and detailed.
   - a)	The description of the listwise ranking loss is not clear. More explanations are needed.
   - b)	Detailed descriptions for obtaining the Pareto ranks (ground truth) for different subnetworks are needed.
   - c)	The detail of the hypervolume improvement should be provided since it’s an important metric for pruning the operations. Besides, the motivation for choosing HVI is unclear. Moreover, the number of sample networks n seems important for the proposed method. How to choose a good n?
 - Estimating architecture latency using a lookup table has been proposed and investigated in FBNet. In my opinion, this is not a good way to evaluate the latency of candidate architectures. In practice, the inference framework (e.g., TensorRT) would perform layer/operation fusion. For instance, TensorRT will fuse conv and batch norm layer into a single layer. In this case, the overall latency of the architecture is not equal to the sum of that of each layers. The reality is often more complicated than the above case. Thus, I think it would be difficult to accurately estimate the latency based on the lookup table.
 - In Eq. 1, the authors state $F_{k’}$ is a set of sub-networks ranked k’. I have no idea what is $F_{k’}$ even aftering carefully reading it. More expalinations should be provide.
 - The ablation studies on “the number of sampled sub-network” and “the number of epochs for Pareto-training” are missing. It would be better for the authors to provide more ablation studies.
 - Some other multi-objective search algorithms [1][2] should also be compared in Tables 2 and 3.
 - The details of the experimental setting in Figures 3 and 5 are unclear. More explanations are needed.
 - The ablation study on the proposed pruning strategy is missing. More experiments are expected.

### Minor issues:

 - In Section 3.2, “Algorithm 1 and figure 1 summarizes summarizes the training procedure” should be “Algorithm 1 and figure 1 summarizes the training procedure”.


### Reference

[1] Nsganetv2: Evolutionary multi-objective surrogate-assisted neural architecture search ECCV 2020.

[2] Ponas: Progressive one-shot neural architecture search for very efficient deployment.



**Summary Of The Paper:**

This paper proposes a supernetwork training strategy for preserving the Pareto ranking between different subnetworks for handware-aware architecture search. To maintain a higher ranking correlation supernetwork, the authors propose to prune the operations based on the optimal Pareto front during the training of the supernetwork. Moreover, the authors define a new metric for better evaluating the architecture in a multi-objective context. Experimental results on three benchmarks demonstrate the effectiveness of the algorithm in some cases. However, the proposed method is not clearly described and the experiments can be further improved. My detailed comments are as follows.

**Summary Of The Review:**

I vote for reject since the proposed method is not clearly described and several key ablations are missing in the paper.

---

> ### Author Response · Authors · 2022-11-19
> **Response to reviewer 4mMw**
>
> We thank the reviewer for thorough analysis of the paper methodology and results and we hope the additional details and explanations make the methodology clearer.
>
> > In the Pareto rank-preserving training section, the details of the proposed method are missing. In other word, after reading this paper, I can not re-implement the proposed method. It would better for the authors to make it more clear and detailed.
>
> To address common concerns of the reviewers, we have structured our first response to clarify and better explain the methodology. All the details in the first response were included in the paper.
>
> > Estimating architecture latency using a lookup table has been proposed and investigated in FBNet. In my opinion, this is not a good way to evaluate the latency of candidate architectures. In practice, the inference framework (e.g., TensorRT) would perform layer/operation fusion. For instance, TensorRT will fuse conv and batch norm layer into a single layer. In this case, the overall latency of the architecture is not equal to the sum of that of each layers. The reality is often more complicated than the above case. Thus, I think it would be difficult to accurately estimate the latency based on the lookup table.
>
> It is true that using inference frameworks and compilers that optimize the architecture graph and include operation fusion, tiling and other optimizations would make the execution non sequential which hinders the use of LUT. However, as this was not our context, we consider the execution to be sequential and we do not use any optimization when deploying our models. We provide experiments in Appendix E to compare LUT to predictor-based runs on NAS-Bench-201 and DARTS. All experiments were run 5 times and the results were averaged. LUT outperforms predictor-based methods on NAS-Bench-201. On DARTS, XGboost is slightly better than LUT (0.94 against 0.91). We do plan to take into account these optimizations in future work but feel that they do not impact the overall results. The methodology still holds whether we use these optimizations or not.
>
> > In Eq. 1, the authors state  Fk  is a set of sub-networks ranked k’. I have no idea what is Fk′ even aftering carefully reading it. More explanations should be provided.
>
> In the revised version, we added figure 2 that explains the Pareto ranking and the different sets of architectures created “Fk”.
>
> > The ablation studies on “the number of sampled sub-network” and “the number of epochs for Pareto-training” are missing. It would be better for the authors to provide more ablation studies. The ablation study on the proposed pruning strategy is missing. More experiments are expected.
>
> Ablation studies were added in the experiments section 5.3.1 and in the appendix C and D.
>
> > The details of the experimental setting in Figures 3 and 5 are unclear. More explanations are needed.
>
> We clarified the experimental settings in section 5.1

---

### Official Review · Reviewer_UeSf · 2022-10-24

**Confidence:** 3
**Correctness:** 3
**Technical Novelty And Significance:** 3
**Empirical Novelty And Significance:** 2
**Recommendation:** 5

**Clarity, Quality, Novelty And Reproducibility:**

Overall, this paper appears to propose a relatively novel take on discovering sub-networks on the Pareto frontier.   However, a lack of related work and detail around the algorithm muddy the presentation, ultimately making it difficult to assess novelty, reproducibility, and applicability.

The end of Sec 1 states that it achieves a "97% near Pareto front approximation" -- it might be straightforward but be very clear if it's a critical metric.

Perhaps the largest omission concerns Algorithm 1.   Section 3.2's "steps" do not seem to accurately reflect the algorithm.  In step 2 "training to solve the task", the bullet only states taking a mini-batch and choosing a subnetwork.  It does not describe training that subnetwork nor give intuition behind that action.   The next section "Pareto-rank training" starts with the sentence "After completing the iteration" but step 2 is part of the iteration?    At least the appendix could spell out the "list wise CE loss" function (and Algorithm 1 could refer to it as opposed to just "Loss_{pr}".    Is \alpha just a single parameter (value) or something else (vector)?   What does "Adjust \alpha" mean?

It's not clear whether you're training the sampled subnetworks from scratch.  I would assume this was the case, since the Pareto training compares the found Pareto ranks (so far) to "ground truth."   However, part of the point of the work was to avoid expensive retraining of the sub-network.   It looks like the benchmarks provided trained networks and in one section you mentioned being able to use those for estimating the Pareto Kendall measure (sec 4.3).   So, does training time (GPU Hours) include that time?

Another gap is that the paper does not investigate the performance / convergence of the Pareto training.   How does the search proceed with N_p?  How many sub-networks should be sampled (n)?   It isn't clear what "Hyper Volume" is or why the reader should care.   It seems to have something to do with completeness of the frontier.   How often will this approach return only rank 1 models?  How often will it return the complete frontier?   How can we give the reader an intuition or proof of that behavior?   I.e., before we compare HW-NAS' final models, it wasn't clear how well the Pareto-training performed.

I'm curious if there is a discussion to be had around the differences of Pareto ranks and performance ranks.   This mechanism will almost always return all models as rank 1 (sec 4.4), but performance ranks will likely have many fewer identical ranks.   Would it be possible to return a set of sub-networks scored rank of 1, but actually they all belong to the second frontier?  When you compute the "ground truth" rank, was it only Pareto ranking the sampled set?  In other words, if the ground truth set was scored in isolation, they would look like rank 1, and the correlation would be artificially high.

The end of 3.1 says we'll only use Pareto ranks, but it would still be nice to repeat in Fig 5 caption.

Related work should take into consideration other pieces of work that have looked at Pareto-based multi-objective formulations, such as
Elsken, Thomas, Metzen, Jan Hendrik, and Hutter, Frank. Efficient multi-objective neural architecture search via lamarckian evolution, ICLR 2019, and Efficient Forward Architecture Search, Hanzhang Hu, John Langford, Rich Caruana, Saurajit Mukherjee Eric Horvitz, Debadeepta Dey, Neurips 2019.    BTW, I got these by looking at prior openreview at ICLR.

**Strength And Weaknesses:**

Positive Points:
+ Multi-objective NAS is an important area, and the proposed method (Pareto-based search training) appears novel
+ Evaluation used 3 benchmarks/search spaces (DARTs, NAS-Bench-201, ProxylessNAS) and additionally compares to FairNAS and a recent one-shot GCN-based technique.
+ Results are promising: optimal sub-networks often meet or improve accuracy/latency while also taking less time to train.

Negative Points
- The paper is written as if optimizing for multiple objectives was a relatively new idea.  But searching for models on the Pareto front has been previously explored in the NAS area.  Related work doesn't describe much of the work in that multi-objective space.
- The description of the algorithm is light on details, specifically the Pareto loss function, the Pareto parameter updates, and how NAs are found.  This makes reproducibility difficult.
- Given the above limited description of the operation of the Pareto-based training, it isn't clear how easy it is to extend to multiple objectives.
- There is little to no evaluation of the Pareto parameter training in isolation to understand its behavior / performance.

**Summary Of The Paper:**

This paper presents a method for multi-objective Neural Architecture Search (NAS), i.e., finding the best NAs based both on task prediction and hardware-based metrics (e.g., latency).  The proposed method (HW-NAS, HW for "hardware") builds on single-stage supernetwork techniques (the union of possible architectural choices for the neural final architecture) by co-training the supernetwork's parameters and a "Pareto" parameter to find (and rank) the sub-models.   They propose to use Pareto-ranks as the primary search objective and NAS evaluation criteria.  They implemented their algorithm and compare the found rank of the models on the Pareto frontier with Pareto rank of the individually trained sub-models for multiple benchmarks.   In addition the trained Pareto parameter allows them to decrease supernetwork training time by pruning sub-networks not be on the Pareto front.    By doing so, they achieve a 97% Pareto front approximation (vs 87%) for the resulting sub-models.

**Summary Of The Review:**

This looks like an interesting and fruitful direction of research - cotraining the supernetwork with a Pareto-based parameter to guide the search.  However, there are presentation, clarity, and background work issues that make it difficult to place the work into context, interpret the results, and understand the algorithm enough for reproducibility.

[Post Author Response] The authors have addressed significant issues regarding the lack of details, including more detailed training steps, specifying the Pareto parameter's loss function, and including new results.   Many of these details should have been present in the original paper. but I've bumped the recommendation to marginal below in light of them.

---

> ### Author Response · Authors · 2022-11-19
> **Response to Reviewer UeSf**
>
> We appreciate the reviewer comments and suggestions.
>
> > The paper is written as if optimizing for multiple objectives was a relatively new idea. But searching for models on the Pareto front has been previously explored in the NAS area. Related work doesn't describe much of the work in that multi-objective space.
>
> We included a related works section relating previous multi-objective hardware-aware neural architecture search as well as the differences between their methodologies and ours. The main issue comes from using multiple estimators (one for each objective). Each estimator brings its share of error and the search becomes biased which results in sub-optimal Pareto front approximation. We enforce the dependency between the objective by making the supernetwork trained to rank using the Pareto ranking metric.
>
> > The description of the algorithm is light on details, specifically the Pareto loss function, the Pareto parameter updates, and how NAs are found. This makes reproducibility difficult.
>
> We added more explanation and formulation to the training section. We refer the reviewer to the common response addressed to all reviewers that explains the algorithm as well as the ground truth computation.
>
> > Another gap is that the paper does not investigate the performance / convergence of the Pareto training. How does the search proceed with N_p? How many sub-networks should be sampled (n)?
>
> Ablation studies and analysis of N_p were performed and included in the Appendix.
>
> > It isn't clear what "Hyper Volume" is or why the reader should care. It seems to have something to do with completeness of the frontier. How often will this approach return only rank 1 models? How often will it return the complete frontier? How can we give the reader an intuition or proof of that behavior? I.e., before we compare HW-NAS' final models, it wasn't clear how well the Pareto-training performed.
>
> We include the hypervolume definition in section 2. The hypervolume indicator measures the effectiveness of the Pareto front approximation. All architectures in the optimal Pareto front approximation are ranked 1. The goal of the training is to keep only architecture with rank 1 in the supernetwork. However, the training sometimes miss-ranks architectures. Compared to other training methodologies, our methods find on average 97% of all optimal points in the Pareto front.
> We include more experiments using the hypervolume in the experiment section.
>
> > I'm curious if there is a discussion to be had around the differences of Pareto ranks and performance ranks. This mechanism will almost always return all models as rank 1 (sec 4.4), but performance ranks will likely have many fewer identical ranks.
>
> “Performance” of architectures in a multi-objective context can be attributed to the task-specific metrics such as accuracy, or hardware efficiency metrics such as latency. Throughout the paper, we argue that using the independent estimations of performances to find the right architecture is sub-optimal. The Pareto ranking, however, includes both objectives using the dominance operation. We also include the definition of the dominance in Related works section 2.

---

### Official Review · Reviewer_qrrY · 2022-10-24

**Confidence:** 4
**Clarity, Quality, Novelty And Reproducibility:** see above
**Correctness:** 3
**Technical Novelty And Significance:** 3
**Empirical Novelty And Significance:** 3
**Recommendation:** 5

**Strength And Weaknesses:**

Strengths:

- The idea is straightforward to implement on existing supernetwork based training schemes, and seems to be effective from the results.
- The methodology is validated on NASBench201, DARTS and ImageNet.

Weakness:

- It is not clear how the second phase of training (argmax(a)) enforces fairness. Justification and implication of unfair sub-network sampling in this stage may be a useful discussion.
- A table showing the pareto approximation effectiveness for each of the tested search spaces would be very useful.
- The current results highlight the fact that PRP-NAS-BL finds a model with low latency, and PRP-NAS-BA finds a model with high accuracy. (As expected.) If possible, comparing with the ground truth optimal architectures in each case may add more value/context to the result.

Questions:

- If argmax(a) selection scheme does not enforce fairness, it is natural to assume that the correlation between argmax(a) and kendall tau rank correlation would increase simply due to the nature of the CELoss minimization, with no bearing on the ground truth. How is the 'Truth' in the CELoss(Score, Truth) formulation calculated?
- If Truth is a list of intermediate accuracies of the sub-networks, then naturally the kendall tau rank correlation will keep increasing, simply due to sampling bias. Comments on this issue would be appreciated.

Minor comments:

- Table 1 (Training hyperparameters) can be moved to the Appendix.
- Fix 3.2 : 'Algorithm 1 and Figure 1 summarizes summarizes..'

**Summary Of The Paper:**

In this paper, the authors introduce a methodology for pareto front sub-network identification from a supernetwork. This is done in a two step process, by first conducting fair subsampling of subnetworks for Nf epochs, followed by Pareto-Rank  training for Np epochs. Sub-networks that are furthest from the pareto front are also dropped to accelerate training. The proposed method achieves state of the art pareto front approximation (97%) in 2 GPU days. A new parameter, $\alpha$ is introduced to measure which operation is critical, and pareto-ranking loss is utilized to adjust the $\alpha$ for different paths. The sum of the selected operations alpha values is utilized as the pareto score and Cross Entropy Loss is taken between the ranking scores and pareto ranks (ground truth) to update $\alpha$.

**Summary Of The Review:**

The paper is interesting but there are some concerns (see above).

---

> ### Author Response · Authors · 2022-11-19
> **Response to Reviewer qrrY**
>
> We thank the reviewer for the valuable comments and remarks. We corrected the minor comments in the current version.
>
> > If argmax(a) selection scheme does not enforce fairness, it is natural to assume that the correlation between argmax(a) and kendall tau rank correlation would increase simply due to the nature of the CELoss minimization, with no bearing on the ground truth. How is the 'Truth' in the CELoss(Score, Truth) formulation calculated?
>
> We understand that our initial description lacks some details to understand how the algorithm is actually learning the operation importance and computing the task-specific loss. We refer the reviewer to the common response addressed to all reviewers that explains the algorithm as well as the ground truth computation.
>
> > A table showing the pareto approximation effectiveness for each of the tested search spaces would be very useful.
>
> We use the hypervolume to evaluate the effectiveness of a Pareto front approximation. Figure 6 of the revised paper shows a comparison of the hypervolume indicator obtained from different multi-objective NAS using different estimators on the DARTS benchmark.
>
> > The current results highlight the fact that PRP-NAS-BL finds a model with low latency, and PRP-NAS-BA finds a model with high accuracy. (As expected.) If possible, comparing with the ground truth optimal architectures in each case may add more value/context to the result.
>
> In the multi-objective context, the result is a Pareto front approximation that contains multiple architectures. We empirically select the optimal architecture presented in table 2 and 3. However, a more accurate metric is the hypervolume. In section 5.2, we show the superposition of our Pareto front approximation to the optimal Pareto front compared to multiple weight-sharing NAS techniques. On NAS-Bench-201, the true and optimal Pareto front is computed with a brute-force algorithm after a few hours which makes it possible to find the normalized hypervolume (hypervolume of our approximation/hypervolume of the true Pareto front). On NAS-Bench-201, we can say that we find 97%, 98% of the optimal Pareto front points on CIFAR-10 and CIFAR-100. However, on DARTS the size of the search space makes it impossible to compute the true Pareto front. We thus show the hypervolume comparison in Figure 6.
>
> > If Truth is a list of intermediate accuracies of the sub-networks, then naturally the kendall tau rank correlation will keep increasing, simply due to sampling bias. Comments on this issue would be appreciated.
>
> The ground truth in our case is a Pareto ranking obtained from the dominance of an architecture. The dominance is measured on multiple objectives. In our case, we focus on accuracy and latency. Note that after the fairness constrained training in step 1, our algorithm becomes unfair towards bad Pareto-ranked architectures (in step 2).

---

### Official Review · Reviewer_Z8Vf · 2022-10-25

**Confidence:** 3
**Correctness:** 3
**Technical Novelty And Significance:** 2
**Empirical Novelty And Significance:** 2
**Recommendation:** 5

**Clarity, Quality, Novelty And Reproducibility:**

Clarity
* This paper is generally well structured. Some details are missing and need to be clarified (see below).

Quality
* The approach is generally well described and is developed based on existing proven method. Some clarifications are needed to justify the correctness (see below).

Novelty
* This work appears to be an extension of superset training to include the proposed Pareto-rank.

Reproducibility
* Cannot be evaluated based on the existing materials.

**Strength And Weaknesses:**

Strength
* Good trade-off between latency and accuracy is achieved using the proposed approach comparing to the SOTA.

Weakness
* It is unclear how the alpha parameter is derived and how does the correctness of accuracy / latency modelling affect the quality of results.

**Summary Of The Paper:**

This paper proposes an hardware-aware NAS on a superset that preserves the Pareto ranking (accuracy and latency). It demonstrates better accuracy-latency tradeoff when compared to SOTA approaches.

**Summary Of The Review:**

* This paper presents an hardware-aware NAS on a superset that consider preservation of Pareto-ranking.
* How is the operation score parameter calculated? Is this the same as the architecture parameter in DARTS? Is this score a good representation of the importance of operation?
* Similarly, the line “Adjust alpha” in Algorithm 1 is not clearly explained.
* The notations in Section 3.2 are slightly confusing, operation score (alpha), Pareto-rank score (alpha), Pareto score (P_s).
* The latency model is based on the summation of layer-wise latency stored in a lookup table. Is this assuming sequential execution of layers in the target hardware? Is this assumption true for the target hardware?
* Are the latency numbers reported in Table 2 and 3 based on real measurement or the latency model?
* Can you provide more comparison to SOTA HW-aware NAS, e.g. BRP-NAS [1] and HELP [2] similar to Table 2 and 3?

Ref:
[1] https://arxiv.org/abs/2007.08668
[2] https://arxiv.org/abs/2106.08630

---

> ### Author Response · Authors · 2022-11-19
> **Response to Reviewer Z8Vf**
>
> We thank the reviewer for the helpful comments and remarks.
>
> > How is the operation score parameter calculated? Is this the same as the architecture parameter in DARTS? Is this score a good representation of the importance of operation? Similarly, the line “Adjust alpha” in Algorithm 1 is not clearly explained.
>
> The operation score alpha is learned using the Pareto training. While in DARTS, this parameter is learned by solving a bi-level optimization problem. We train the alpha score using a hinge loss that compares the number of times an operation i in layer j is selected in the top N sub-networks of the Pareto front (ground truth) to the number of times it is selected when argmax(alpha) = i. Compared to DARTS, this score is a good representation of the importance of an operation as it is directly related to the ground truth pareto front for each sampled sub-networks.
> In algorithm 1, we updated the line “Adjust alpha” to “Update alpha by gradients” which better reflects the training of alpha.
>
> > The notations in Section 3.2 are slightly confusing, operation score (alpha), Pareto-rank score (alpha), Pareto score (P_s).
>
> This was a writing mistake, it is fixed in the current version.
>
> > The latency model is based on the summation of layer-wise latency stored in a lookup table. Is this assuming sequential execution of layers in the target hardware? Is this assumption true for the target hardware?
>
> We added a comparison of different latency estimation techniques to validate the use of LUT in Appendix E. Note that we deploy our architectures directly without using any optimization strategy including tensorRT operator fusion optimization.
>
> > Are the latency numbers reported in Table 2 and 3 based on real measurement or the latency model?
>
> The latency numbers reported in the experimentation section and in the additional results section in the Appendix are all real measurements.
>
> > Can you provide more comparison to SOTA HW-aware NAS, e.g. BRP-NAS [1] and HELP [2] similar to Table 2 and 3?
>
> BRP-NAS and HELP both use independent surrogate models for accuracy and latency. We added a comparison with these techniques in terms of search time and hypervolume in figure 6.

---

> > ### Comment · Reviewer_Z8Vf · 2022-12-06
> > **Thanks for the clarification**
> >
> > I appreciate the authors' response and paper revision, which adequately addressed many comments of the reviewers. I have updated the score accordingly.

---

### Decision · Program_Chairs · 2023-01-20

**Decision:**

Reject

**Justification For Why Not Higher Score:**

The clarity of the paper does not meet the standard of a research paper, as it leaves out tons of details to reproduce the method and the experimental results.

**Justification For Why Not Lower Score:**

N/A

**Metareview: Summary, Strengths And Weaknesses:**

This paper proposes a supernetwork-based approach to hardware-aware neural architecture search (HW-NAS), which consists of subnetworks that form the Pareto front in terms of accuracy-latency tradeoff across a set of hardware platforms. This is done by first training the supernetwork with strict fairness across operations (using FairNAS), and pruning the operations to obtain subnetworks that have higher Pareto-rankings. The experimental validation of the method on multiple hardware platforms against an extensive set of HW-NAS, or NAS baselines show that the proposed method can obtain architectures that have better Pareto-rankings compared to baselines (better accuracy for the same latency, or better latency for the same accuracy).

The reviewers in general agree that the proposed supernet-based approach to HW-NAS is well-motivated, as it is addressing an important challenge, and find the method as promising as it obtains architectures with good trade-offs between the accuracy and latency, against an extensive set of baselines. However, they were also concerned about lack of novelty since existing works on HW-NAS also consider searching for Pareto-optimal architectures. Also, more importantly, they were concerned with very unclear descriptions of the methods, which makes the work irreproducible, which was pointed by all reviewers and were the main reason they unanimously recommended rejection.

The authors provide short responses to the reviewers during the author-reviewer discussion period, which mostly focus on clarifications, and partly revised the paper. However, clarity on the method and the experimental evaluation part still subpar, making it extremely difficult to reproduce the work, given that there is no code or the supernetworks provided. I suggest the authors to completely rewrite the paper with clear descriptions of both the methods and the experimental evaluation part. Another, minor concern on my side is the lack of experimental validation on an exhaustive set of hardware devices, since the method seems to have been validated on only few devices. Since the hardware devices may exhibit largely different latency-generating behaviors depending on the platforms, there should be a more exhaustive experimental validation of the method over a diverse set of devices.